

# The neuropsychological profile of professional action video game players

Julie Justine Benoit[1,2], Eugenie Roudaia[3], Taylor Johnson[4], Trevor Love[4] and Jocelyn Faubert[2]

[1] Department of Psychology, Université de Montréal, Montréal, Québec, Canada
[2] Faubert Lab, École d'Optométrie, Université de Montréal, Montréal, Québec, Canada
[3] Rotman Research Institute, Toronto, Ontario, Canada
[4] Infinite Esports and Entertainment, Frisco, TX, United-States

Corresponding author
Julie Justine Benoit,
juliejustine_benoit@hotmail.com

## ABSTRACT

In the past 20 years, there has been growing research interest in the association between video games and cognition. Although many studies have found that video game players are better than non-players in multiple cognitive domains, other studies failed to replicate these results. Until now, the vast majority of studies defined video game players based on the number of hours an individual spent playing video games, with relatively few studies focusing on video game expertise using performance criteria. In the current study, we sought to examine whether individuals who play video games at a professional level in the esports industry differ from amateur video game players in their cognitive and learning abilities. We assessed 14 video game players who play in a competitive league (Professional) and 16 casual video game players (Amateur) on set of standard neuropsychological tests evaluating processing speed, attention, memory, executive functions, and manual dexterity. We also examined participants' ability to improve performance on a dynamic visual attention task that required tracking multiple objects in three-dimensions (3D-MOT) over five sessions. Professional players showed the largest performance advantage relative to Amateur players in a test of visual spatial memory (Spatial Span), with more modest benefits in a test of selective and sustained attention (d2 Test of Attention), and test of auditory working memory (Digit Span). Professional players also showed better speed thresholds in the 3D-MOT task overall, but the rate of improvement with training did not differ in the two groups. Future longitudinal studies of elite video game experts are required to determine whether the observed performance benefits of professional gamers may be due to their greater engagement in video game play, or due to pre-existing differences that promote achievement of high performance in action video games.

## INTRODUCTION

There were more than two billion video gamers worldwide in 2016 and this number is projected to increase to 2.7 billion by 2021 (*Statista, 2020*). In the United States, the gamer population consists of more than 150 million individuals, representing a 17.7 billion dollar market (*Bediou et al., 2018*; *SpillGames, 2013*; *Statista, 2017*). There is a wide variety of video game genres, including action, real-time strategy, fighting, adventure, role playing,

and racing games. Action video games (AVGs) such as *Call of Duty*, Grand Theft Auto, *Halo, Fallout 4, Fortnite*, and *Overwatch,* are among the most popular types of video games in the United States (*Statista, 2019*). More recently, we have witnessed the emergence of the eSports industry, in which video gamers compete individually or in teams in national and international competitions. More than 117 schools in the United States offer competitive eSports programs and many professional leagues are experiencing growing audiences and revenues (*NACE, 2020*; *Newzoo, 2020*).

The rise in popularity of video games in the last 20 years has led to a surge of research examining their impact on the mind and brain, with a special focus on AVGs. Although AVGs differ from one another, they all share four characteristics: a fast pace (moving objects, time constraints), a high perceptual load, a high degree of distraction, and a requirement for constant switching between focused and distributed states of attention (*Bediou et al., 2018*). AVGs are also highly engaging and intrinsically motivating activities, making them attractive and popular (*Powers & Brooks, 2014*). First-person shooter (FPS) games, in which the player has an egocentric view through his or her avatar's eyes, have been the focus of many studies, as they were suspected to be the most likely genre of AVGs to influence cognition due to their high engagement of sensory, perceptual, and cognitive functions (*Spence & Feng, 2010*). Although AVGs are the most studied, other types of games, like real-time strategy games, have also been shown to affect cognition (*Glass, Maddox & Love, 2013*). Cross-sectional studies comparing habitual players of AVGs and non-players have reported that, as a group, habitual players of AVGs perform better than non-players in multiple cognitive domains, including selective attention (*Castel, Pratt & Drummond, 2005*; *Dye, Green & Bavelier, 2009a*; *Dye, Green & Bavelier, 2009b*; *Green & Bavelier, 2003*; *Green & Bavelier, 2006a*; *Green & Bavelier, 2006b*), speed of processing (*Castel, Pratt & Drummond, 2005*; *Dye, Green & Bavelier, 2009a*; *Dye, Green & Bavelier, 2009b*), executive functions (*Andrews & Murphy, 2006*; *Colzato et al., 2010*) and working memory (*Colzato et al., 2013*). These cross-sectional, observational studies have been reinforced by several intervention studies that have demonstrated an improvement in the same cognitive domains in non-players following training with AVGs (*Feng, Spence & Pratt, 2007*; *Powers & Brooks, 2014*; *Spence et al., 2009*). A recent meta-analysis of 82 studies focusing on AVGs concluded that AVGs are associated with improved cognitive function in general, with the most robust effects seen in the domains of spatial cognition, top-down attention, and perception; medium effects seen in multitasking and task-switching; and only weak effects seen in inhibition and verbal cognition (*Bediou et al., 2018*). Nevertheless, several other studies have failed to find benefits of video gaming using similar methodologies (*Cain, Landau & Shimamura, 2012*; *Irons, Remington & McLean, 2011*; *Murphy & Spencer, 2009*).

The vast majority of studies examining the effects of AVGs on perceptual and cognitive function studied video game players who were defined based on a criterion amount of time they spent engaging in video game play over the past 6–12 months, with only a few studies also characterizing players based on their level of performance in a game. *Latham, Patston & Tippett (2013)* argued that the lack of consideration of individual differences in the level of performance in video games, in addition to experience, likely contributes to the heterogeneous results in the literature and limits our understanding of video game

expertise. One category of experts that excels relative to others are elites. In the broadest sense, experts can be described as individuals who acquire knowledge or abilities in a specific domain such as a profession, hobby, sport, or game, by devoting a substantial amount of time to that activity (*Chi, Glaser & Farr, 1988*; *Ericsson & Towne, 2013*; *Farrington-Darby & Wilson, 2006*). Elites are experts who achieve a high level of performance in their domain relative to others. In the context of sports for example, athletes who play in professional leagues or who rank highly in international competitions are considered elites (*Swann, Moran & Piggott, 2015*). In video games, elites are those players who consistently achieve high rankings or who are selected to participate in professional leagues. While practice is necessary to achieve high levels of performance, the amount of practice alone is not sufficient, as an individual can practice a lot and acquire knowledge in a specific area without ever becoming an elite (*Ericsson & Towne, 2013*).

Research on expertise has sought to understand whether there are certain characteristics that enable individuals to achieve high levels of performance in various domains (*Chi, Glaser & Farr, 1988*). In sports, research has focused on evaluating perceptual-cognitive capacities, which refer to the ability to identify and amass information to combine with actual knowledge in order to select and execute the appropriate response (*Chi, Glaser & Farr, 1988*; *Mann et al., 2007*). The ability to extract information rapidly is a crucial element of high-level competitive sports, as athletes must direct their attention to the relevant aspects of the wide and dense visual scene in order to make fast decisions (*Mann et al., 2007*). A meta-analysis of athletes across various sports reported that elite athletes are indeed better able to extract pertinent information in the visual scene in the context of their sport, while also showing a different pattern of eye movements and visual search strategies relative to non-elite athletes (*Mann et al., 2007*). Perceptual-cognitive capacities may also be important in videogaming, where a player must also select and extract relevant information to keep track of his or her enemies while anticipating their actions and deciding the best strategy to reach his or her objective.

More broadly, it has been suggested that individuals who achieve high levels of expertise also show enhanced performance across a wide range of core cognitive or perceptual domains in general, outside the specific context of their expertise. To continue with the example of sports, several recent studies have found that elite athletes outperform non-athletes in cognitive tests evaluating attention, multitasking, working memory, and processing speed, with group effects ranging from small to medium effect sizes (*Faubert, 2013*; *Scharfen & Memmert, 2019*; *Vaughan & Laborde, 2020*; *Voss et al., 2010*). Professional athletes also showed faster improvement in performance on a three-dimensional multiple-object-tracking task (3D-MOT) as a function of training, compared to high-level amateurs and to non-athletes (*Faubert, 2013*; *Faubert & Sidebottom, 2012*). The 3D-MOT task determines the speed at which the participant can track a subset of identical moving objects in a three-dimensional space over several seconds. The maximum speed at which a given number of targets can be tracked shows large individual variability, improves with training (*Faubert, 2013*; *Legault, Allard & Faubert, 2013*; *Parsons et al., 2016*; *Tullo, Faubert & Bertone, 2018*), and is associated with better decision making in sports (*Romeas, Guldner & Faubert, 2016*). The finding that the improvement in 3D-MOT performance
with practice was greater in professionals may point to generally enhanced learning abilities within the context of any dynamic visual scene. In sum, the studies above suggest that the outstanding performance achieved by elite athletes may be associated with enhanced abilities in a range of cognitive domains, including attention, processing speed, working memory, and learning abilities.

To our knowledge, no existing studies have examined whether professional action video game players also exhibit enhanced cognitive abilities relative to amateur players. While growing evidence supports a link between playing action video games and cognitive ability (*Bediou et al., 2018*), further research is needed to better understand the nature of this association. Studying cognitive abilities of professional video gamers can shed more light on this relationship, by providing clues as to how expertise is developed, the mechanisms at play, and how best to support and improve it (*Farrington-Darby & Wilson, 2006*). Specifically, we can address the following question: does the outstanding performance of professional video game players stem from enhanced cognitive abilities in certain domains, or is it largely due to a greater expertise within the context of the game?

To address this gap, this study aimed to characterize the cognitive and learning abilities of high-performance action video game players recruited amongst the Houston Outlaws, a professional team in the Overwatch league™. We used a selection of standardized neuropsychological tests to evaluate cognitive abilities, and trained participants on the 3D-MOT task to assess their abilities to learn a novel, dynamic perceptual-cognitive task. We hypothesized that professional players would perform better than amateur players on neuropsychological tasks that evaluate attention, processing speed, executive functions, working memory, and visuo-spatial manipulation, as these aspects of cognition have been found to be more developed in habitual video game players when compared with non-players (*Bediou et al., 2018*). We also hypothesized that professional players would show faster learning rates on 3D-MOT compared with amateur players, as was observed with expert athletes (*Faubert, 2013*).

## MATERIALS & METHODS

### Participants

The experimental protocol was evaluated and approved by the Comité d'Éthique de la Recherche en Santé of Université de Montréal (18-009-CERES-D). Fourteen participants (all men, right-handed) were recruited amongst competitive players in the Overwatch League™ for the Houston Outlaws (Professional group). These participants are considered elite video gamers, because they have achieved the necessary performance level to enter a professional league. They reported daily FPS video game usage in the last 6 months and were ranked as Grandmaster or Top 500/Pro in the game. As a comparison group, we recruited habitual video game players (Amateur group) through online advertisements targeted at undergraduate students at the Université de Montréal. Knowing that the video game category and type may be important influencing factors (*Dobrowolski et al., 2015*), participants in the Amateur group needed have played more than 5 h per week of FPS during the last 6 months. They also could not have previously participated in organized video

**Table 1  Demographic and video game experience information for the two groups.**

| | Professionals (n = 14) M (SD) | Amateurs (n = 16) M (SD) | Group comparison |
|---|---|---|---|
| Age (years) | 23.66 (2.44) | 25.31(3.77) | $t(25.9) = -1.44, p = 0.16$ |
| Gender ratio (male: female) | 14:0 | 12:4 | N / A |
| BDI-II score | 6.85 (5.27) | 4.93 (4.62) | $t(24.1) = 1.01, p = 0.32$ |
| Age started playing video games (years) | 6.93 (2.81) | 6.75 (2.62) | $t(26.8) = 0.18, p = 0.86$ |
| Average FPS gaming per week in past six months (h) | 55.79 (16.72) | 9.47 (3.48) | $t(14) = 10.18, p = 0.00$ |
| Most frequent game | Overwatch (14) | Counter Strike (7), Call of Duty (3), Overwatch (3), Rainbow 6 (1), Player Unknown's Battlegrounds (1), Battlefield 4 (1), | N / A |

**Notes.**

BDI-II, Beck Depression Inventory (II) score; FPS, first person shooter.

game competitions or played more than 20 h per week in the past 6 months. This exclusion criterion was used to ensure that the Amateur group was homogeneous and similar to previous studies in the range of hours of game play per week (*Green & Bavelier, 2003*; *Green & Bavelier, 2006a*; *Green & Bavelier, 2006b*; *Karle, Watter & Shedden, 2010*; *Colzato et al., 2013*). A total of 20 participants (3 left-handed, 4 female) were enrolled in the Amateur group. Two participants were then excluded for playing more than 20 h per week or for having taken part in competitions, and two more were excluded due to inadequate testing environments, resulting in a sample of 16 participants in the Amateur group. Demographic and video game experience characteristics of the two groups are summarized in Table 1. All participants were screened for depressive symptoms using the Beck Depression Inventory II (BDI-II) (*Wang & Gorenstein, 2013*); no participants were excluded based on their BDI-II scores (exclusion > 20). Participants had normal or corrected-to-normal vision and were free of visual, neurological, musculoskeletal, cardiovascular and vestibular impairments, as assessed by self-report. Handedness was also self-reported. All participants gave their verbal and written informed consent to participate after receiving verbal and written information about the study. They were not paid for their participation.

## Neuropsychological measures

Table 2 summarizes the eight neuropsychological tests that were used in this study.

The d2 Test of Attention (*Brickenkam & Zillmer, 1998*) was selected to evaluate selective and sustained attention skills, as well as speed of processing. In this test, participants are presented with a sheet of paper containing 14 lines of 47 items each. The items are either a ''p'' or a ''d'' with one to four dashes placed alone or in pairs below and above the letter. Participants are given 20 s per line to cross out all the items containing a ''d'' with two dashes. The outcome measures include the total number of items processed (TN), percent of errors of omission and commission (%E), number of correct items (TN-E),

**Table 2  Neuropsychological assessments.**

| D2 – Test of attention | Selective attention, sustained attention, concentration |
|---|---|
| WMS-III –Spatial Span | Visual short-term and working memory |
| WAIS-IV –Visual Puzzles | Perceptual reasoning and perceptual manipulation |
| WAIS-IV –Coding | Speed of processing |
| WAIS-IV –Digit Span | Auditory short-term and working memory |
| D-KEFS –Towers | Executive function, planning |
| D-KEFS –Color-Word Interference | Executive function, inhibition and task switching |
| Grooved Pegboard | Eye-hand coordination, dexterity |

concentration performance (CP), and variability in performance across lines (fluctuation ratio, FR).

The WAIS-IV Coding test was selected to evaluate visual processing speed (*Wechsler, 2011*). In this task, participants are required to code a series of numbers using symbols shown on a key at the top of the page, similar to the Digit Symbol Substitution Test. The total number of items that are coded in two minutes is recorded.

The WAIS-IV Visual Puzzles was selected to evaluate visual reasoning and the ability to manipulate visual information (*Wechsler, 2011*). This task requires the participant to decide which three of six puzzle pieces combine together to reconstruct a larger puzzle within a limited time. The number of successfully completed puzzles is scored.

The WAIS-IV Digit Span test (*Wechsler, 2011*) was selected to evaluate auditory working memory (aWM) and short-term memory (aSTM). The test requires participants to listen to a series of digits that are read out loud and to recite them back in the same order (Forward subtest), in backwards order (Backward subtest), or in increasing numerical order (Sequencing subtest). The total number of correctly reported sequences is scored (*Weiss et al., 2010*).

The Wechsler Memory Scale-III (WMS-III) Spatial Span test (*Kessels et al., 2008*) was selected to evaluate visual working memory (vWM) and visual short-term memory (vSTM). In this test, participants are shown nine cubes placed randomly on a board. The examiner taps a number of cubes in a sequence and participants have to reproduce this sequence either in the order presented (Forward subtest) or in backwards order (Backward subtest). The total number of successful sequences on each subtest is scored.

Two tests from the Delis Kaplan Executive Function System (D-KEFS) were selected to evaluate executive function (*Delis, Kaplan & Kramer, 2001*). The D-KEFS Tower test evaluates problem solving and planning. The task requires moving five disks across three pegs to build a tower in the fewest number of moves possible. Performance is scored by combining the total achievement score, which is the sum of achievement points for all the administered items, and the move accuracy ratio score, which assesses the efficiency with which the participant constructed the towers.

The D-KEFS Colour-Word Interference test is a version of the Stroop test that evaluates inhibition and cognitive flexibility. In the Inhibition condition, participants are required to name the colour of the ink of a series of words that spell a name of a different colour. In the Flexibility condition, participants are required to read the words that are outlined by a

rectangle, but to name the ink colour of the other words that are not outlined. The time required to complete a set of words is recorded (*Strauss, Sherman & Spreen, 2006*).

Finally, the Grooved Pegboard test (Lafayette Model 32025) was selected to evaluate hand-eye coordination and dexterity (*Strauss, Sherman & Spreen, 2006*). This test requires participants to pick up small metal pegs that have a key on one end and insert them into randomly oriented slots on the board by rotating the pegs into the correct position. The test is performed with each hand separately and the time required to insert twenty-five pegs into the slots is recorded.

### Three-dimensional multiple objects tracking (3D-MOT)

The 3D-MOT sessions were conducted in a quiet room using a fully immersive environment with a Fove$^{TM}$ virtual reality device. The head-mounted display had a resolution of 2,560 × 1,440 pixels and covered a maximal visual field of 45 degrees. The 3D-MOT task was delivered using a Dell Inspiration 15 700 Gaming Series computer.

The 3D-MOT was developed as an optimal training procedure to improve mental abilities critical for processing dynamic scenes, such as those encountered during sports or video gaming (*Faubert & Sidebottom, 2012*). The procedure adapts the well-known multiple object tracking task (*Pylyshyn & Storm, 1988*), in which participants track several moving targets among distractors, by expanding it to cover a large span of the visual field in three dimensions using stereoscopic presentation, and by varying the speed of the moving objects on every trial to determine a speed threshold (*Faubert & Sidebottom, 2012*).

During the 3D-MOT task (Fig. 1), participants were required to track four of eight spheres that moved within a cube delineated by light grey walls with their attention only, without eye movements. A green fixation square was presented in the center of the cube and participants were asked to maintain fixation on the square throughout the tracking phase. Each trial began with the presentation of all eight yellow spheres positioned at random locations within the cube for two seconds. The four target spheres were then highlighted by changing to a red colour with a white halo for two seconds. Once the targets returned to the yellow colour for once second, all eight spheres began to move along linear paths in random directions within the cube, changing directions when colliding with each other or with the walls. After eight seconds, the spheres stopped moving and were labeled with numbers 1 through 8. The participant was then prompted to verbally identify the target spheres. After their choice was entered, the correct targets were revealed for two seconds to provide the participant feedback. The next trial began shortly afterwards. The speed of all the spheres varied across trials according to a one-up, one-down staircase procedure to estimate the speed required to track all four targets correctly 50% of the time (*Levitt, 1971*). The speed increased by 0.05 log units if the participant correctly identified all targets and decreased by the same amount if the participant missed at least one target. The staircase procedure was interrupted after 20 trials and the speed threshold was estimated using the geometric mean of the speeds for the last four reversals. A complete staircase procedure lasted approximately eight minutes.
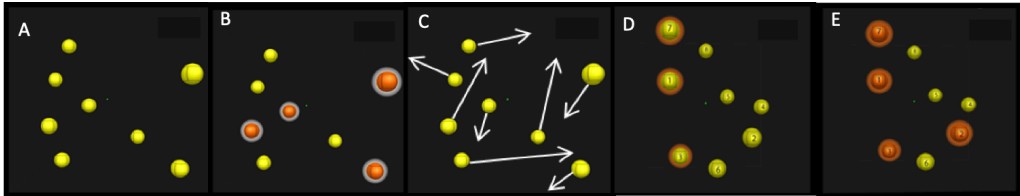

**Figure 1 Five stages of a trial in the 3D-MOT task.** (A) Presentation phase, in which eight spheres appear in random locations in a 3D space. (B) Indexing phase, in which four spheres are cued as targets with a 1 second colour change to red and the appearance of a halo. (C) Movement phase, in which all spheres move for eight seconds in random directions, crisscrossing and bouncing off of each other and the walls of the 3D space and the participant is asked to keep track of the targets, while fixating at the central fixation point. (D) Identification phase, in which the immobilized spheres are numbered 1 –8 and the participant is asked to identify the four targets. (E) Feedback phase, in which the target spheres are highlighted in red and a star is shown if all four targets were selected correctly.

## Procedure

Participants in the Professional group were tested at their headquarters located in Burbank, California, and in Frisco, Texas. Participants in the Amateur group were tested in their homes in Montreal. The same experimenter administered all the tests. In the first visit, participants first completed a demographics and video game experience questionnaire. Then, participants completed the paper and pencil neuropsychological tests in their preferred language (English or French), which lasted approximately one hour. Participants then completed the 3D-MOT task three times. Subsequent 3D-MOT training sessions were completed in four additional sessions, which were spaced apart by a minimum of 6 h and a maximum of 72 h.

## Analysis

Two participants in the Professional group did not have high proficiency in English or French, so they were not tested on the Digit Span and Colour Word Interference tests, since performance in these tests depends on language proficiency. One of these participants also did not complete the Tower test, due to difficulties with understanding instructions. Data for the 3D-MOT training sessions were available for 27 of 30 participants, because three participants in the Amateur group did not complete the five 3D-MOT sessions due to travel restrictions, so their data were excluded from these analyses. Additionally, data from 17 out of 405 blocks across seven participants were missing due to a technical error and were treated as missing at random.

Statistical analyses were performed in SPSS and in the statistical computing environment R (R Core Team, 2015). First, we compared performance on all the neuropsychological measures in the two groups using independent Welch's $t$-tests, as the variance in the two groups was unequal for some outcome measures (e.g., Spatial Span total score, Levene's test $p < 0.01$). Hedges' $g$ was calculated to provide a measure of effect size using the effsize package in R (*Torchiano, 2016*). To evaluate global group differences in neuropsychological measures and to compare the contribution of different outcome measures to the group difference, we conducted a descriptive discriminant analysis (*Brown & Wicker, 2000*; *Smith,*

*Lamb & Henson, 2020*). Additionally, bivariate Spearman's correlations were calculated to examine associations among neuropsychological outcome measures.

To examine the effects of learning on speed thresholds in both groups, we conducted a linear mixed effects analysis using the lme4, pbkrtest, and lmerTest packages in R (*Baayen, Davidson & Bates, 2008*; *Bates et al., 2014*; *Kuznetsova, Brockhoff & Christensen, 2017*; *Luke, 2017*; *R Core Team, 2020*). The model's outcome variable was the speed threshold, with fixed effects of group, the logarithm of the block number, and their interaction. As random effects, we fit a maximal random effects structure that included by-subject intercepts and by-subject slopes for block (lmer(speed ~Group * log2(block) + (1+log2(block) | subject)). This analysis is equivalent to fitting a logarithmic learning curve separately for each participant and then evaluating the effect of group on overall speed thresholds and learning rate. *P*-values for fixed effects were obtained using F-tests with the Kenward-Roger approximation for degrees of freedom (*Halekoh & Højsgaard, 2014*).

# RESULTS

Table 1 summarises the demographic information and video game experience of participants in the groups. The two groups were well matched in age (Professionals: $M = 23.66$, SD $= 2.44$, Amateurs: $M = 25.31$, SD $= 3.77$, t(25.9) $= -1.44$, $p = 0.16$) and in the average age at which participants started playing video games (Professionals: $M = 6.93$, SD $= 2.81$, Amateurs: $M = 6.75$, SD $= 2.62$, t(26.8) $= 0.18$, $p = 0.86$). Participants in the Professional group devoted approximately five times more time to FPS games than those in the Amateur group in the last six months (Professionals: $M = 55.79$ h, SD $= 16.72$, Amateurs: $M = 9.47$ h, SD $= 3.48$, t(14) $= 10.18$, $p < 0.001$). There was no evidence for any differences in depression symptoms between both groups (Professionals: $M = 6.85$, SD $= 5.27$, Amateurs: $M = 4.93$, SD $= 4.62$, t(24.1) $= 1.01$, $p = 0.32$).

## Neuropsychological assessments

Figure 2 shows the outcome measures from the eight neuropsychological tests for the two groups. Table 3 presents their summary statistics and the results of univariate, between-group statistical analyses for each measure.

In the d2 Test of Attention, Professional players processed, on average, a greater number of stimuli than those in the Amateur group (TN: t(28) $= 2.41$, $p = 0.02$, $g = 0.85$), while maintaining a similar error rate (E%: t(27.3) $= 0.70$, p $= 0.$49, $g = 0.25$). When accounting for the error rate, Professional players processed a greater number of stimuli correctly (TN-E: t(28) $= 2.3$; $p = 0.03$; $g = 0.81$), but their Concentration Performance score did not differ significantly from the Amateur group (CP: $t(26.4) = 1.31$; $p = 0.20$; $g = 0.47$). The Fluctuation Rate, which measures the consistency of performance throughout the task, was lower in the Professional group (FR: t(28)$=-2,14$; $p = 0.04$; $g = -0.75$), indicating better sustained attention.

In the WAIS-IV Spatial Span task, the Professional group showed better performance than the Amateurs on the Total score (t(25.5) $= 3.52$, $p < 0.001$, $g = 1.27$), with a large effect size. Analyzing each subscale separately revealed a large, reliable effect in the Forward subscale, (t(28) $= 3.85$; $p = 0.001$, $g = 1.36$), but no reliable difference in the Backward

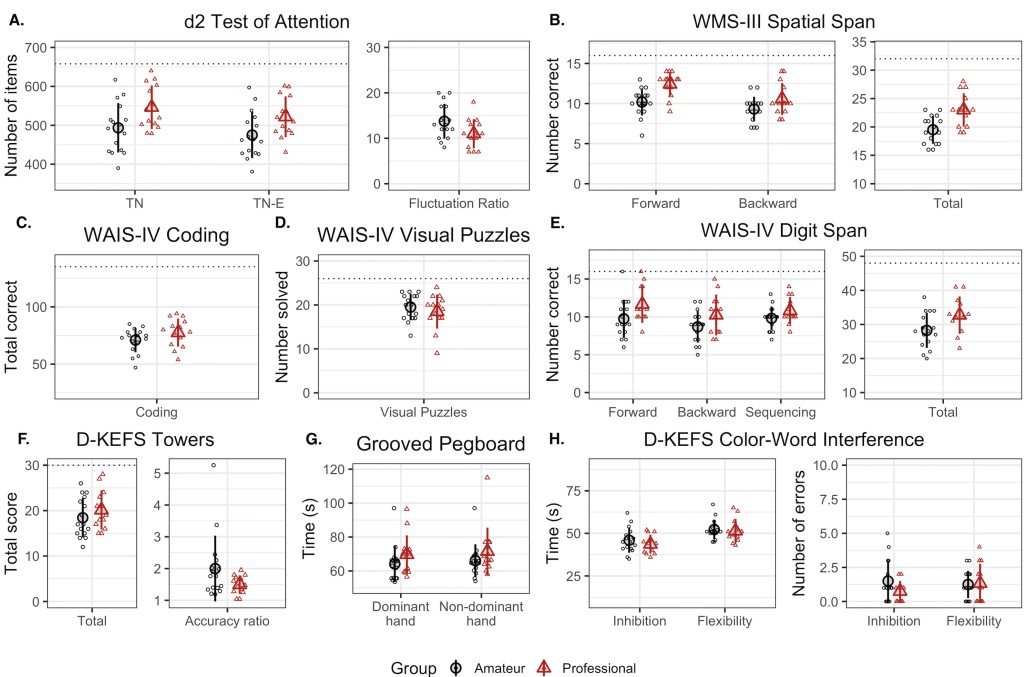

**Figure 2 Results of neuropsychological tests.** Large symbols show the mean group score and small symbols show individual participant data for the Amateur (black circles) and Professional (red triangles) groups. Error bars represent 1 SD of the mean. The dashed lines represent the maximum score possible for each variable. (A) d2 Test of Attention. (B) WMS-III Spatial Span. (C) WAIS-IV Coding. (D) WAIS-IV Visual Puzzles. (E) WAIS-IV Digit Span. (F) D-KEFS Towers. (G) Grooved Pegboard. (H) D-KEFS Color-Word Interference.

subscale ($t(24.4) = 1.92, p = 0.07, g = 0.7$). In the WAIS-IV Digit Span task, the Professional group also showed better performance on the Total score ($t(23) = 2.20; p = 0.04, g = 0.82$), with a smaller effect size than in the Spatial Span test. Scores for the Digit Span Forward, Backward, and Sequencing subscales were not reliably different in the two groups ($p = 0.06, 0.11, 0.13$, respectively).

In the WAIS-IV Coding test, the average score in the Professional group was numerically higher than that of the Amateur group, but this group difference was not reliable ($t(26) = 1.54, p = 0.14, g = 0.55$). There was no evidence for a difference in performance between the two groups on tests of executive function, D-KEFS Towers (ps > 0.08) and D-KEFS Colour-Word Interference (ps > 0.36), nor on the Visual Puzzles test (ps = 0.44), which evaluates perceptual reasoning. There was also no evidence for any difference between groups in the Grooved Pegboard test (ps > 0.17), which evaluates hand-eye coordination and manual dexterity.

Next, we used a multivariate analysis approach to probe for global differences between the two groups across all neuropsychological assessments. To ward against multicollinearity, we selected one measure per task and conducted a descriptive discriminant analysis (DDA; *Smith, Lamb & Henson, 2020*) on this subset of outcome measures, The measures included were d2 TN-E, Spatial Span Total, Digit Span Total, Grooved Peg for the dominant hand,

**Table 3 Average raw scores for all neuropsychological tests, difference scores, and univariate inferential statistic.**

| Measure | Professionals (n = 14) M (SD) | Amateurs (n = 16) M (SD) | Difference Δ [95% CI] | Welch t-test t(df) | p | Effect size Hedges' g g [95% CI] |
|---|---|---|---|---|---|---|
| D2 Test of Attention | | | | | | |
| –TN | 546.29 (56.14) | 493.56 (63.49) | 52.73 [7.98, 97.47] | t(28) = 2.41 | **0.02** | 0.85 [0.09, 1.61] |
| –E% | 4.41 (2.98) | 3.66 (2.91) | 0.75 [−1.46, 2.96] | t(27.3) = 0.7 | 0.49 | 0.25 [−0.48, 0.98] |
| –TN-E | 521.86 (52.48) | 474.94 (59.31) | 46.92 [5.11, 88.73] | t(28) = 2.3 | **0.03** | 0.81 [0.05, 1.57] |
| –CP | 208.79 (31.62) | 194.38 (28.41) | 14.41 [−8.26, 37.08] | t(26.4) = 1.31 | 0.20 | 0.47 [−0.27, 1.21] |
| –FR | 11 (3.21) | 13.75 (3.84) | −2.75 [−5.39, −0.11] | t(28) = −2.14 | **0.04** | −0.75 [-1.51 ,0] |
| WAIS-IV Coding | 77.5 (12.25) | 71.0 (10.61) | 6.5 [−2.16, 15.16] | t(26) = 1.54 | 0.14 | 0.55 [−0.19, 1.3] |
| WMS-III Spatial Span | | | | | | |
| –Total | 23 (2.94) | 19.5 (2.45) | 3.5 [1.45, 5.55] | t(25.5) = 3.52 | **0.002** | 1.27 [0.47, 2.07] |
| –Forward | 12.43 (1.5) | 10.19 (1.68) | 2.24 [1.05, 3.43] | t(28) = 3.85 | **0.001** | 1.36 [0.55, 2.17] |
| –Backward | 10.57 (1.99) | 9.31 (1.54) | 1.26 [−0.09, 2.61] | t(24.4) = 1.92 | 0.07 | 0.7 [−0.06, 1.45] |
| WAIS-IV Digit Span[a] | | | | | | |
| –Total | 32.75 (5.5) | 28.25 (5.17) | 4.5 [0.27, 8.73] | t(23) = 2.2 | **0.04** | 0.82 [0.03, 1.62] |
| –Forward | 11.67 (2.46) | 9.75 (2.52) | 1.92 [−0.04, 3.88] | t(24.1) = 2.02 | 0.06 | 0.75 [−0.04, 1.53] |
| –Backward | 10.25 (2.7) | 8.69 (2.09) | 1.56 [−0.39, 3.52] | t(20.1) = 1.67 | 0.11 | 0.64 [−0.14, 1.42] |
| –Sequencing | 10.83 (1.8) | 9.81 (1.56) | 1.02 [−0.33, 2.37] | t(21.8) = 1.57 | 0.13 | 0.6 [−0.18, 1.37] |
| WAIS-IV | | | | | | |
| Visual Puzzles | 18.5 (3.88) | 19.5 (3.01) | −1.00 [−3.64, 1.64] | t(24.4) = −0.78 | 0.44 | −0.28 [−1.02, 0.45] |
| D-KEFS Towers[b] | | | | | | |
| –Total | 20.15 (4.32) | 18.44 (4.32) | 1.71 [−1.6, 5.03] | t(25.8) = 1.06 | 0.30 | 0.39 [−0.37, 1.14] |
| –Accuracy Ratio | 1.49 (0.29) | 2 (1.04) | −0.51 [−1.08, 0.06] | t(17.9) = −1.86 | 0.08 | −0.62 [−1.38, 0.15] |
| D-KEFS Color-Word | | | | | | |
| –Inhibition | 43.67 (5.58) | 45.94 (7.35) | −2.27 [−7.3, 2.75] | t(26) = −0.93 | 0.36 | −0.33 [−1.1, 0.44] |
| –Flexibility | 51.42 (6.97) | 52.25 (5.7) | −0.83 [−5.96, 4.3] | t(20.9) = −0.34 | 0.74 | −0.13 [−0.89, 0.63] |
| Grooved Pegboard | | | | | | |
| –DH | 69.73 (11.26) | 64.1 (10.68) | 5.63 [−2.62, 13.89] | t(27) = 1.4 | 0.17 | 0.5 [−0.24, 1.24] |
| –NDH | 71.58 (13.95) | 66.12 (9.65) | 5.46 [−3.73, 14.66] | t(22.7) = 1.23 | 0.23 | 0.45 [−0.29, 1.19] |

**Notes.**

Means and standard deviations (SD) of the raw scores of neuropsychological tests for Professional and Amateur video gamer participants.

a: n = 12 in the Professional group; b: n = 13 in the Professional group.

WAIS, Weschler Adult Intelligence Scale (IV); WMS, Weschler Memory Scale (III); TN, total number of items processed,; TN-E, total number of items processed minus the errors,; E%, percentage of errors; CP, concentration performance; FR, fluctuation rate (difference between the line with the minimum and maximum number of items processed).; DH, Dominant hand; NDH, Non-dominant hand.

Bolded p-values are < an alpha level of 0.05 (uncorrected).

Color-Word Inhibition score, and total scores for Coding, Visual Puzzles, and Towers. The largest bivariate correlation across these measures was −0.55, indicating that the variables were not multicollinear. The DDA analysis calculates a linear composite of the outcome variables that best separates the two groups. The canonical correlation, $R_c$, between the composite and Group was 0.67, with a corresponding $R_c^2 = 0.45$, which is the variance accounted for Group in the composite scores. This difference was not statistically significant, Wilk's $\Lambda = 0.54$, $F(8, 19) = 1.97$, $p = 0.11$. Examining the standardized discriminant function coefficients revealed that the Spatial Span Total score made the

**Table 4 Results of the linear mixed effects model fit to speed thresholds in the 3D-MOT task.**

| Fixed Effect | β | 95% CI | $t$ (df) | $p$ | Random effect | St. Dev. | Corr |
|---|---|---|---|---|---|---|---|
| Intercept | 0.59 | [0.42, 0.76] | $t(25.8) = 6.72$ | <0.001 | Subj Intercept | 0.24 | |
| Group | 0.28 | [0.05, 0.52] | $t(24.8) = 2.35$ | 0.03 | Subj log2(block) | 0.10 | 0.06 |
| log2(block) | 0.15 | [0.13, 0.29] | $t(26.1) = 5.21$ | <0.001 | Residual | 0.28 | |
| Group: log2(block) | 0.06 | [−0.05, 0.17] | $t(24.8) = 1.10$ | 0.28 | | | |

**Notes.**
The proportion of the variance explained by fixed and random factors, conditional $R^2$, is 0.68; the proportion of the variance explained by the fixed factors alone, marginal $R^2$, is 0.29.

largest contribution to the composite score, with a coefficient of −0.29 and $r^2 = 0.62$.
Table 4 provides the full results of this analysis.

### 3D-MOT training

Figure 3 shows the average speed thresholds for tracking four targets in the 3D-MOT task
as a function of block number for each group. As can be seen, the Professional group
showed higher thresholds than the Amateur group, indicating better ability to track
multiple targets among distractors. Thresholds in both groups increased as a function of
block number, reflecting improvements in task performance as a function of training, with
a similar rate of improvement in both groups. These observations were confirmed by a
linear mixed effects analysis (see Table 5), which revealed a main effect of Group, F(1,
24.78) = 5.54, $p = 0.03$, with the Professional group having higher thresholds, β = 0.28,
95% CI [0.05–0.52]. There was also a main effect of block, F(1, 24.78) = 76.97, $p < 0.001$,
with thresholds increasing by β = 0.14, 95% CI [0.09–0.20] for every doubling of the block
number. The interaction between Group and log2(block) was not statistically significant,
F(1, 24.78) = 1.23, $p = 0.28$, providing no evidence that the learning rates differed between
the two groups.

### Association between attention, working memory, and short-term memory

Previous studies have indicated that visual working memory (WM) and selective attention
are related both on a behavioral and neuroanatomical level (*Awh & Jonides, 2001*; *Gazzaley
& Nobre, 2012*). It has been argued that selective attention is crucial for effectively filtering
irrelevant information at the encoding phase (*Vogel, McCollough & Machizawa, 2005*).
Because WM has a limited capacity, attention is required to appropriately select the
relevant information to encode in WM to avoid unnecessary clutter (*Ma, Husain & Bays,
2014*; *Myers, Stokes & Nobre, 2017*). Selective attention may also be involved in maintaining
information activated in WM (*Awh & Jonides, 2001*; *Awh, Vogel & Oh, 2006*). Many studies
also report an association between WM and STM (*Conway et al., 2002*; *Engle et al., 1999*;
*Kail & Hall, 2001*), as STM is often theorized as a subcomponent of WM (*Cowan, 1998*;
*Engle et al., 1999*; *Kail & Hall, 2001*).

Figure 4 shows the results of bivariate Spearman's correlations among attention and
processing speed outcome measures (d2 Test of Attention TN-E, WAIS-IV Coding, and
the average speed threshold for the 3D-MOT task at baseline), and among auditory and

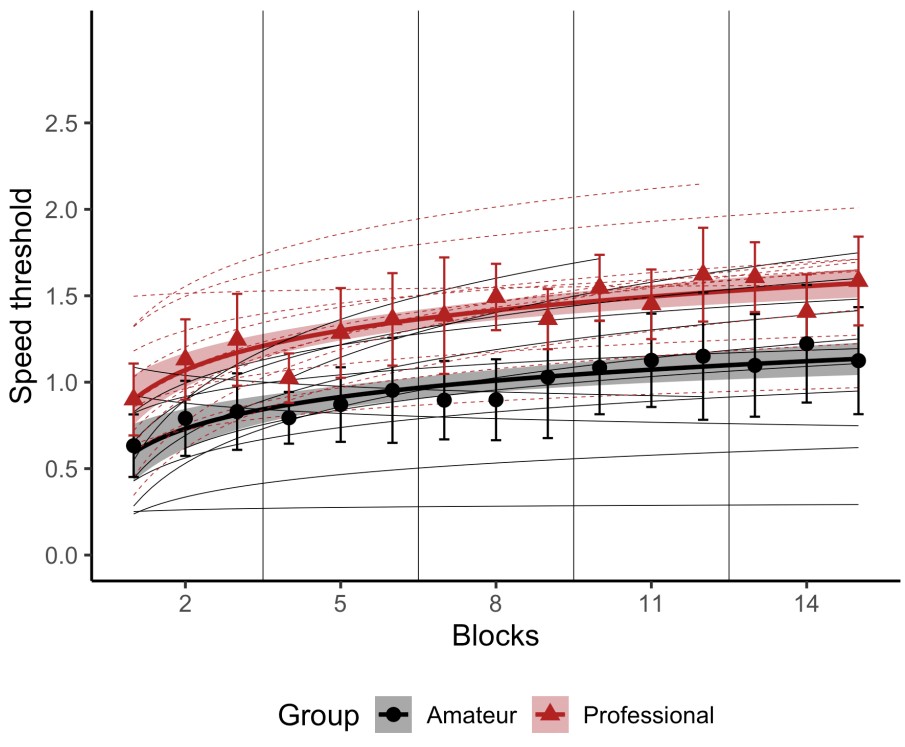

**Figure 3** **3D-MOT task training data.** Average speed thresholds for the Professional (red triangles) and Amateur (black circles) group are plotted as a function of block number. Participants completed three blocks in each of five sessions. The error bars represent standard deviations of the mean. Thin lines show logarithmic regression fits to individual participants' data (dashed red for Professionals, solid black for Amateurs), with thick lines showing the group fits. Speed is expressed in arbitrary speed units.

visual WM and STM outcome measures (Spatial Span Forward and Backward subscales, Digit Span Forward, Backward, and Sequencing subscales). We only examined correlations for both groups together, as the sample sizes in each group were not sufficient to provide stable correlation estimates (*Schönbrodt & Perugini, 2013*). As can be seen, d2 TN-E scores were correlated with Spatial Span Backward subscale ($r_s = 0.50$, $p = 0.01$), and with the WAIS-IV Coding ($r_s = 0.45$, $p = 0.01$), but not with Spatial Span Forward ($r_s = 0.21$, $p = 0.26$). Thus, our measure of selective attention correlated with measures of vWM and processing speed, but not with vSTM. The d2 TN-E score also correlated with the Digit Span Sequencing subscale ($r_s = 0.51$, $p = 0.01$), but not with the Forward or Backward subscales. The Spatial Span Forward and Backward subscales were moderately correlated with each other ($r_s = 0.43$, $p = 0.02$). Performance on the 3D-MOT task showed a strong correlation with the Spatial Stan Forward ($r_s = 0.68$, $p < 0.001$) and Backward ($r_s = 0.69$, $p < 0.001$) subscales, indicating that these two tasks index similar cognitive capacities. Finally, the Spatial Span Forward was the only variable that correlated with the average weekly hours of video game play ($r_s = 0.45$, $p = 0.12$). Keeping in mind that the Professional and Amateur groups showed a large difference in their average weekly hours of video game play, this

**Table 5  Standardized coefficient, structure coefficients, and groups centroids.**

| Outcome variable | Standardized coefficient | $r_s$ | $r_s^2$ |
|---|---|---|---|
| WAIS-IV –Digit Span Total | −0.15 | −0.59 | 0.35 |
| WMS-III –Spatial Span Total | −0.79 | −0.79 | 0.62 |
| D-KEFS –Tower Total Score | −0.25 | −0.32 | 0.10 |
| WAIS-IV –Visual Puzzles | 0.39 | 0.24 | 0.06 |
| D-KEFS –Color word –Inhibition | 0.02 | 0.26 | 0.07 |
| D2 - TN | −0.28 | −0.61 | 0.37 |
| WAIS-IV - Coding | −0.12 | −0.35 | 0.12 |
| Grooved Pegboard - Dominant Hand | −0.37 | −0.27 | 0.07 |
| Group | Centroids [95% CI] | Cohen's d [95% CI] | |
| Professional | −1.1 [−1.68, −0.524] | −1.84 [−2.82, −0.83] | |
| Amateur | 0.76 [0.275, 1.25] | | |

**Notes.**

WAIS, Weschler Adult Intelligence Scale (IV); WMS, Weschler Memory Scale (III); D-KEFS, Deli- Kaplan Executive Function System; TN, total number of items processed.

positive association is consistent with the previously-reported difference in Spatial Span performance in the two groups.

## DISCUSSION

To gain a better understanding of the cognitive determinants of expertise in action video games, the current study compared the performance of professional action video gamers with that of video gamers who play similar FPS games non-competitively on a set of eight neuropsychological tests, and on their ability to improve their performance on a multiple objects tracking task. The results revealed that Professionals performed better than Amateurs on some measures in the Spatial Span, Digit Span, d2 Test of Attention, and 3D-MOT tasks. These results indicate that high performance in FPS video games is associated with enhanced abilities in visual and auditory short-term memory, selective and sustained attention, and visual spatial attention. There was no evidence for any group differences in performance on tasks that evaluated executive functions, perceptual manipulation, or manual dexterity. Furthermore, both groups showed similar capacity to improve their performance in the 3D-MOT task with training. Given the cross-sectional, observational nature of this group comparison, this study cannot speak to the causality of the differences in cognitive performance and video game expertise. The presence of differences between groups may either indicate that certain cognitive abilities are conducive to achieving high performance in FPS games, or that the greater amount of experience with FPS games in the professional group lead to improved performance in the above-mentioned abilities.

The results from the d2 Test of Attention indicate that Professional players may have better selective and sustained attention than Amateur players. This finding is consistent with previous studies that reported attentional benefits in non video-game players following practice with action video games (*Belchior et al., 2013*; *Green & Bavelier, 2003*; *Green & Bavelier, 2006a*; *Green & Bavelier, 2006b*; *Spence et al., 2009*). It is also consistent with

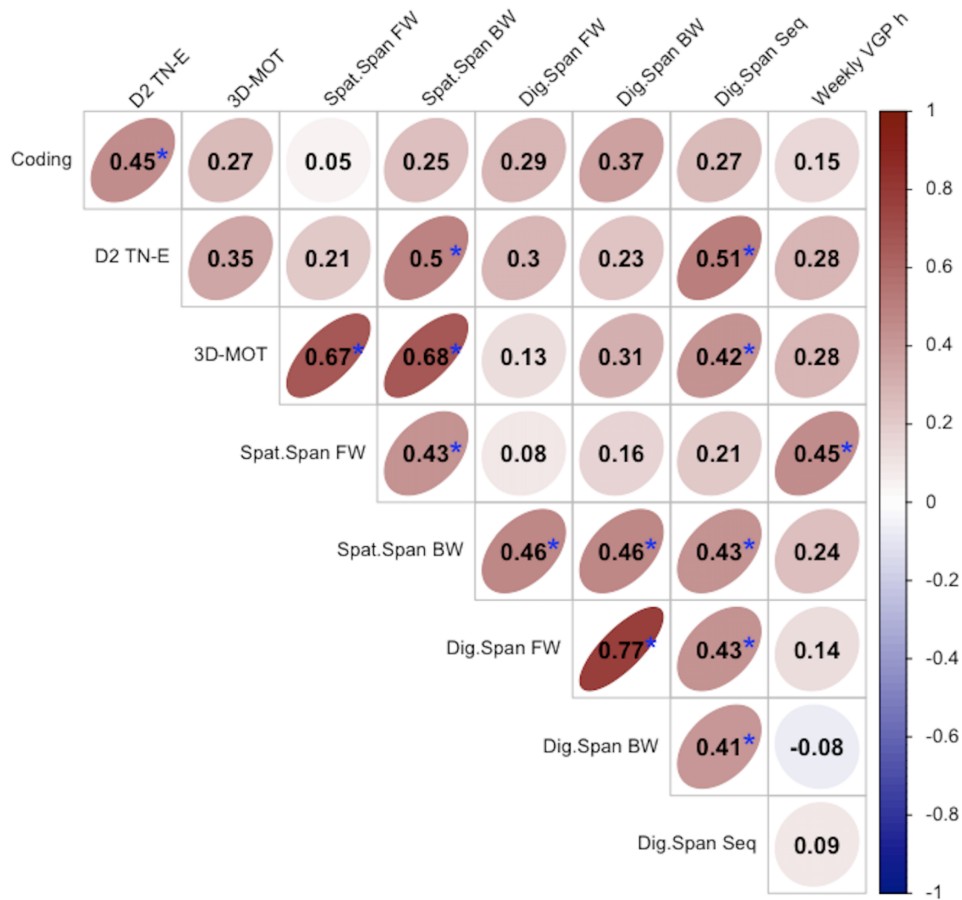

**Figure 4  Bivariate Spearman's correlations for measures of attention and working memory for all participants.** Blue star indicates *p* < 0.05, uncorrected for multiple comparisons. VGP h: average hours of action video game play per week.

results of a meta-analysis that found strong evidence for a robust effect of action video games on attention using similar tasks (*Bediou et al., 2018*). Some authors have raised the possibility that AVG players show higher performance in attention tasks because they employ a more optimal visual search strategy (*Clark, Fleck & Mitroff, 2011*). We believe that differences in search strategies were minimized in the d2 test, because participants were instructed to search line by line without the possibility of going back. Furthermore, although Professional players processed, on average, more stimuli than Amateurs, they maintained the same accuracy rate, suggesting no apparent trade-offs between speed and accuracy. These results suggest that Professional players displayed enhanced visual selective attention compared to Amateurs. Furthermore, the performance of Professional players was more stable than Amateurs' in terms of items processed per line, suggesting better capacities in sustained attention. Sustained attention is an important ability in the context of professional videogaming, where players often need to train for many hours consecutively. While previous research has demonstrated that AVG players hold an advantage over non-players on attention (*Castel, Pratt & Drummond, 2005*; *Green &*

*Bavelier, 2006a*; *Green & Bavelier, 2006b*), the current results indicate that selective and sustained attention are further enhanced in elite video game experts. Thus, taken together, these results suggest that attention capacities could be influenced by AVGs.

Previous studies that used change detection tasks with simple and complex stimuli have reported better vSTM capacities in action video game players (*Blacker & Curby, 2013*; *Wilms, Petersen & Vangkilde, 2013*). In the present study, vSTM was indexed using the Spatial Span Forward subscale, which showed the greatest benefit in performance in the Professional group. Previous researchers have proposed that improvements in vSTM could be linked to the enhancement of visual selective attention that enables to select task-relevant and ignore task-irrelevant information (*Bavelier et al., 2012*). In the context of our study, the bivariate correlation did not show a statistically significant association between our attention and vSTM measures, suggesting that the benefits in vSTM seen here are not strongly related to selective attention abilities.

While previous studies mainly examined the effects of video games on visual cognition, a few studies have also noted benefits of video games on performance in auditory or multisensory tasks. For example, adult video game players showed better temporal processing of multisensory stimuli than non-players (*Donohue, Woldorff & Mitroff, 2010*) and training with action video games was associated with improvements in reading and phonological short-term memory in children with dyslexia (*Franceschini & Bertoni, 2019*; *Franceschini et al., 2013*). In the present study, there was some evidence for a benefit in auditory memory in the Digit Span test for the Professional group, although there was no evidence of a difference in the Forward subscale that indexes auditory short-term memory. While experience with video games may impact auditory and multisensory performance, we do not find evidence that auditory short-term memory is a characteristic of video game expertise.

The present study suggests a link between the level of video gaming expertise and the ability to perform an abstract dynamic task. The 3D-MOT task strongly engages several attention and mental skills; performing this task well requires selective, dynamic, distributed, and sustained attention skills (*Faubert, 2013*). Professional video game players performed better than Amateurs on the 3D-MOT task across the 5 training sessions, suggesting that video game expertise is also related to perceptive-cognitive ability. This is consistent with the idea that Professionals players must be efficient at extracting meaningful information from a visual scene in order to anticipate and make good decisions. The 3D-MOT task results are consistent with the enhancement of motion perception highlighted by other studies that employed the multiple objects tracking or dots motion tasks (*Boot et al., 2008*; *Green & Bavelier, 2006a*; *Green & Bavelier, 2006b*; *Hutchinson & Stocks, 2013*). These results are also consistent with the growing evidence that action video game experience has an impact on the dorsal pathway (*Chopin, Bediou & Bavelier, 2019*). The dorsal pathway is a network involved in spatial working memory. It specializes in capturing dynamic spatial and temporal relationships between multiple items (*Kravitz et al., 2013*), and is engaged during multiple object tracking tasks (*Blumberg, Peterson & Parasuraman, 2015*; *Howe et al., 2009*). The dorsal pathway is often referred to as the 'where/how' pathway, as it is involved in localizing and guiding motor action (*Chopin, Bediou & Bavelier, 2019*).

Furthermore, the professional and casual video game players in the current study showed similar learning rates across the five sessions, in contrast to other studies, where professional sports athletes showed enhanced abilities to improve their 3D-MOT performance with training relative to amateur athletes (*Faubert, 2013*).

While the effect in the multiple object tracking task is consistent with differences in the dorsal pathway, there were no group differences in the Grooved Pegboard task, which relies on action planning and visuomotor coordination that also depends on the dorsal pathway. Previous studies found that players have better hand-eye coordination than non-players, but this benefit was not associated with the amount of time spent engaging in the games (*Griffith et al., 1983*). Action video games were also shown to improve visuomotor control in an intervention study (*Li, Chen & Chen, 2016*). Together with the current findings, this suggests that the benefits of action video games on motor dexterity and visuomotor coordination are limited, in that extended practice does not lead to larger benefits.

The current results also do not provide evidence for any differences in working memory associated with professional gaming, as the average performance on the Backward subscales of the Spatial Span and Digit Span tasks did not differ between groups. As mentioned earlier, previous studies have emphasized that WM, which involves maintaining elements active and quickly accessible, is intricately linked with the concept of selective attention, both on a behavioral and anatomical level (*Awh & Jonides, 2001*; *Engle, 2002*; *Hitch et al., 2018*). Supporting this overlap between the two constructs, we observed positive associations between outcome measures of vWM, selective attention, and vSTM. Nevertheless, while there was some evidence for enhanced selective attention in professional players, performance in tasks relying on auditory or visual working memory were similar in both groups.

The present results also do not provide evidence for any differences in executive functions between Professional players and Amateurs, as measured by the inhibition and flexibility subscales of the Stroop task. Mental flexibility has been cited as one of the strongest enhancements in real-time strategy games (*Basak et al., 2008*). The meta-analysis in AVGs conducted by *Bediou et al. (2018)* suggested a medium size effect for flexibility, and only a weak effect for inhibition. Given that the current study is only well suited to detect large effects, our findings are consistent with previous results. There is also no evidence of any group differences in visual planning skills or visual reasoning. These results are consistent with the results of Boot and colleagues (*2008*), who found no association between video game practice and planning skills, using the Tower of London task (*Tunstall, 1999*).

Previous studies have found that video game players are faster than non-video game players in reaction time tasks (*Castel, Pratt & Drummond, 2005*; *Dye, Green & Bavelier, 2009a*; *Dye, Green & Bavelier, 2009b*). However, the current study found no reliable difference in performance in the main visual processing speed task (WAIS-IV Coding), nor in the Grooved Peg task, which relies on rapid manual dexterity and hand-eye coordination. That said, the TN measure of the d2 Test of Attention showed that Professional players process a greater number of stimuli in a given time than Amateurs, suggesting that video game expertise is related to a better ability to process simple visual information. The different results provided by the d2 and Coding tests address a question raised by *Dye,*

*Green & Bavelier (2009a)* and *Dye, Green & Bavelier (2009b)*, who wondered whether the advantages of video gaming are restricted to tasks involving only binary responses, or if they can generalize to more complex tasks with multiple response alternatives. The Coding task requires participants to quickly code a series of items using the correct symbol among a total of nine codes. Thus, in addition to speed of processing, good performance in the Coding task requires attention, motor speed, visuo-perceptual abilities, and dexterity to write the appropriate symbols (*Jaeger, 2018*). In contrast, the participant response in the d2 test involves a choice between one of two options: to mark or not to mark the symbol. The two-choice response in the d2 test is closer to that used in the literature demonstrating the benefits of video gaming on processing speed (*Dye, Green & Bavelier, 2009a*; *Dye, Green & Bavelier, 2009b*). The current findings may suggest that video game expertise is related to a better ability to process simple visual information, rather than providing a global benefit in processing speed in more complex tasks.

By better understanding Professional videogaming, we can provide clues as to how this expertise and the corresponding performance best develops, and therefore learn how to support it (*Farrington-Darby & Wilson, 2006*). In order to effectively study video game expertise, it is crucial to define it appropriately. The definitions of video gamers in previous studies ignored potential differences between video game players who had played 5 h per week over the previous 6 months, and those who had played more than 20 h per week over the previous 10 years. Moreover, the assumption that recent video gaming experience reflects expertise could be mistaken (*Latham, Patston & Tippett, 2013*). Failure to consider expertise in terms of performance may also contribute to some of the mixed results in the literature. By applying the same performance criteria to video gaming research, differences between professional and amateur players allowed us to determine the factors that potentially underlie high-level performance, such as attention, STM, and perceptive-cognitive ability.

The current study has serval limitations. First, due to our limited sample size, the current study had low power to detect small or medium effects (with alpha = 0.05, power = 0.80). Thus, we should remain careful in our interpretation of null results, as differences might exist but failed to be detected. Furthermore, the gender of the study participants may have acted as a confounding variable since our Professional players group was composed only of men, whereas our Amateur group had four women, although women's results did not differ from men's within the same group. Finally, the experimenter who administered the measures was not blind to the participant group, which may have introduced some bias.

## CONCLUSIONS

In summary, this study was the first to examine the cognitive basis of elite performance in action video game players. Our results revealed that elite players show the greatest performance advantage in tests of visual spatial short-term memory and of visual attention. Furthermore, professional action video gamers showed a better ability to track multiple objects within a complex and dynamic scene than amateur players, but both groups showed similar rates of improvement in the task with training. Further research is needed to clarify

whether the observed differences in cognitive abilities emerge as a result of intense practice in action video games, or whether certain cognitive profiles are beneficial for achieving high-level performance in video gaming.

## ACKNOWLEDGEMENTS

We would like to acknowledge Russel Smith, Matt Men, and Stephen Suttle for their support during data collection.

### Funding

This work was supported by Natural Sciences and Engineering Research Council of Canada discovery grant. The funders had no role in study design, data collection and analysis, decision to publish, or preparation of the manuscript.

### Grant Disclosures

The following grant information was disclosed by the authors:
Natural Sciences and Engineering Research Council of Canada discovery grant.

### Competing Interests

One of the authors, Jocelyn Faubert, is the director of Faubert Lab at the University of Montreal and he is also the Chief Science Officer of Cognisens Athletics Inc. who produces the commercial version of the NeuroTracker used in this study. In this capacity, he holds shares in the company. Taylor Johnson and Trevor Love are employed by Infinite Esports and Entertainments.

### Author Contributions

- Julie Justine Benoit conceived and designed the experiments, performed the experiments, analyzed the data, prepared figures and/or tables, and approved the final draft.
- Eugenie Roudaia analyzed the data, prepared figures and/or tables, authored or reviewed drafts of the paper, grammar and writing, and approved the final draft.
- Taylor Johnson and Trevor Love performed the experiments, authored or reviewed drafts of the paper, subject recruiting, and approved the final draft.
- Jocelyn Faubert conceived and designed the experiments, authored or reviewed drafts of the paper, and approved the final draft.

### Human Ethics

The following information was supplied relating to ethical approvals (i.e., approving body and any reference numbers):
   Comité d'Éthique de la Recherche en Santé of Université de Montréal approved this research (18-009-CERES-D).

### Data Availability

   The raw data are available in the Supplemental File.

## Supplemental Information

Supplemental information for this article can be found online at http://dx.doi.org/10.7717/peerj.10211#supplemental-information.

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
