# Peer review of "The neuropsychological profile of professional action video game players"

_PeerJ, doi:10.7717/peerj.10211_

## Round 0.1 · original submission · Major Revisions

This is in general a well-written and interesting paper that presents an interesting result (if somewhat preliminary, due to a necessarily small sample size). As noted by the reviewers, there are numerous small English-language errors, and the paper would benefit from a copyediting pass by a native English speaker. These do not detract from the understandability of the article, however. Similarly, there are some issues that should be addressed in the discussion/introduction, and some issues with the statistical analysis, as noted by the reviewers. Although I am marking this as "Major Revision required," due to the possibility that the additional reanalyses will meaningfully change the results, I think that most of the other comments can be addressed in a fairly straightforward manner. In preparing the ms for publication, it would also help to increase font size on the axes labels in Figure 2 and, if possible, directly label the groups on the data curves instead of requiring the reader to refer to the caption (failing this, a legend could be provided on the graph). Additionally, please provide error bars on Figure 2 to indicate within-group variability.

Reviewer 1 ·

Basic reporting

The article is generally written clearly and professional English is used throughout. In the annotated document I have highlighted some instances where the language was not clear or was ambiguous.

For example, in the abstract the phraseology 'do not make unanimity' is a bit awkward. I believe Statista (2018; on line 42) is using 'AVG' in different sense from the video game literature which makes the opening paragraph a bit difficult to follow. And so on. These should not take long to correct.

The article is well referenced however, there are two pockets of literature which need to be included. First, its acknowledged at the start of the paper that researchers have found null results in the video game literature but none of these studies are cited. Second, the authors emphasise the relation between video game play and cognitive abilities. And given that I think the authors think that the direction of the relation runs from video game play to cognitive abilities it would have been good to cite some targeted training studies with healthy populations.

I have included some examples of authors in both groups in the annotated document.

Experimental design

There are some missing details in the methods section. For example, the reader is not told what the selection criteria the casual video game group had to respect was. Nor are we told how many were excluded for failing those criteria (or the BDI-II inventory for that matter). Was handedness self-reported or assessed using something like the EHI? And so on.

There are a number of other minor queries of this nature in the annotated document.

Validity of the findings

It would have been better to report exact p-values when possible, and report standard deviations rather than standard errors (unless the authors have some reason for doing otherwise). There are a number of other minor queries in the annotated document. For the most part this section is fine.

Currently, the discussion really only acknowledges that the current results are consistent with elite video game players being self selected at the very end. Up to that point the authors describe video game play being 'related' to certain capacities but 'related' here is ambiguous. While there are training studies which show that no video game players who play video games show abilities in the direction of causal video game players, there is currently no training studies that show that causal video game players matching elite video game players. This is no problem but it does raise interesting questions about the use and testing of elite players (supposedly it would be to identify target capacities for enhancement) and what the possible limits of training might be. Given that this is a cross sectional study there are limitations on what can be concluded about the relationship between video gaming demographics and cognitive capacities.

Issues surrounding this point come up a number of times in the discussion and there are number of comments and suggestions in the annotated document that I hope will help.

Additional comments

This was an enjoyable paper which makes a novel contribution to the literature through presenting findings of actual professional video game players (Houston Outlaws). Very few research groups have access to such demonstrable experts video game players and so this research will be on interest to a wide readership.

I have attached an annotated manuscript with questions, queries, concerns and suggestions which I had while reading through the manuscript. I hope this document assist the authors with making revisions to the manuscript. The changes are not substantial but should hopefully make the paper much stronger.

Annotated reviews are not available for download in order to protect the identity of reviewers who chose to remain anonymous.

·

Basic reporting

no comment

Experimental design

no comment

Validity of the findings

no comment

Additional comments

In the present article the Authors investigated the cognitive differences in two groups of video game players, i.e. elite n=14 vs. casual n=16. The findings indicate that elite video game players, in comparison to casual video game players, show higher performance in visual attention and working memory, as well as in the learning of 3D multiple-object-tracking task, suggesting a better functioning of dorsal ("When" and Where") "Action" pathway.

The article is interesting and it is a very good paper for Peer J.

Here some constructive (major and minor) comments:


Main comments

1. The relationship between selective attention and working memory could be better describe considering also the relevant reviews such as:
a. Edward Awh, John Jonides (2001). Overlapping mechanisms of attention and spatial working memory. Trends in Cognitive Sciences, Vol. 5, Issue 3, p119–126
b. Adam Gazzaley, Anna C. Nobre (2011). Top-down modulation: bridging selective attention and working memory. Trends in Cognitive Sciences, Vol. 16, Issue 2, p129–135
c. Nicholas E. Myers, Mark G. Stokes, Anna C. Nobre (2017). Prioritizing Information during Working Memory: Beyond Sustained Internal Attention. Trends in Cognitive Sciences, Vol. 21, Issue 6, p449–461

2. Only group analysis are reported. I suggest to explore also individual data. How many EVGPs perform better than the control group? For example, the Authors could use a cut off of 1 DS.
In addition, in order to explore better the possible relationship between selective attention and working memory in visual domain (main comment 1), a simple bivariate correlation analysis could be used. The same for the possible relation between working memory in visual and auditory domains (see main point 4).

3. A multivariate analysis between the two groups could be use to investigate the global results of the complex neuropsychological evaluation.

4. The multi-sensory effects of FPS action video games (i.e., WAIS Digit Span result) could be better considered also reporting recent results from action video game training in children with neurodevelopmental disorders such as developmental dyslexia (see Franceschini et al., 2013 Curr Biol, 2017 Sci Rep; Franceschini & Bertoni, 2019 NeuroPsychologia).

5. The interesting result of learning during 3D-MOT task could suggest the relevant role of FPS action video games mainly on dorsal-action pathway (but see the null results on visuo-motor skills also reported in the present article).

Minor comments
-pg. 8/32 line 66 "spatial cognition" is reported both as "robust impact" (line 65) as well as "less substantial" (line 68).
-pg 16/32 line 251 and 252 please check and report complete statistical results (reported in Table 3).
-Table 1 is equal to Table 2. Probably the Table 1 should report Neuropsychological tests.

---

## Round 0.2 · Minor Revisions

You will see that Reviewer 1 feels that you were generally responsive to all suggestions/criticisms, and I agree that the paper is greatly improved as a result, and worthy of publication.

Reviewer 1 also noted some remaining (very) minor issues. Because there are quite a few of these, some of which rise beyond the level of simple copyediting errors, I am issuing a decision of Minor Revisions *only* to give you the opportunity to consider adopting these suggestions and to perform one last proofreading pass at this stage.

Please then resubmit the article, at which point I will likely be able to issue an Accept decision with out the need for sending out for re-review. To be clear, you may choose which suggestions you choose to adopt/address, but I believe you will be well-served to consider most or all of them.

A few additional minor points that I noticed:
- line 271 "allowed" should presumably be "awarded"
- line 256 et seq. "last" should be "lasted"
- line 280 "a... questionnaires" should either "a ... questionnaire" (singular noun) or delete "a" and make it simply "...questionnaires"
- lines 748, 810 appear to have missing information

Again, a careful proofread at this final stage should take care of these and any other minor errors. (If you prepare a rebuttal letter, you do *not* need to explicitly list individual proofing corrections.)

I'll look forward to receiving (and accepting) a final revised version.

Reviewer 1 ·

Basic reporting

Minor Comments:

Line 58
"between focused and distributed state of attention" missing 'a' before focused?

Line 61
FPS games are introduced as a sub-category of AVGs. However, all the previous examples of AVGs on Line 49-50 are FPS games. If AVGs are meant to be a more general category, then it might be worth giving some examples earlier of AVGs which are not FPS games.

Line 65
"must" is meant to be 'most'.

Line 90
"The concept of elite experts is useful when considering difference in performance amongst experts."? Unclear sentence.

Paragraph beginning Line 119
While I get the comparison being drawn elite athletes I think that it would be worth making clear that this is just being used as a paradigm case of expertise and its associated wide-ranging benefits. After all, the same point being made here with elite athletes could also be made using elite musicians (for example) who also show similar wide-ranging benefits (including to the visuospatial domain).

Line 138
Not sure what cognitive expertise means in this context. Is is something over and above the previously listed enhanced cognitive abilities?

Line 141
Is "distinct" the right word? 'Enhanced'?

Line 146
Change "extraordinary". Perhaps just use 'enhanced' due to previous usage.

Line 156
"Will" should be 'would'

Line 158
"were" should be 'have been found'

Section beginning Line 191
For clarity start a new paragraph with each measure.

Line 214
Incomplete brackets

Line 231
The Grooved pegboard test description could be clearer.

Line 479
Missing 'a'

Line 491
Sentence beginning "Those studies..." is unclear.

Line 496
It might be worth making absolutely clear that while it might not be a characteristic of expertise it is open that it might be a characteristic of video game experience (or exposure of some sorts).

Line 508
Could be stated more clearly. The behavioural results observed here are not evidence that action video games impact on the dorsal pathway. They are simply consistent with the hypothesis that they do impact on the dorsal pathway.

Line 517
"athletes" is repeated

Line 519
Any thoughts why there might be such a difference between athletes and video game players on the 3D-MOT?

Line 552
"planification" better as 'planning'?

Line 561
What was the question raised by Dye et al. (2009)?

Line 569
Delete "Matthew WG"

Line 576
Beginning of the sentence could be stated more clearly.

Experimental design

Minor Comments:

Line 179
What was the reason for excluding participants who played more than 20 hours per week? Given the focus on expertise, then so long as these participants were not competitive (the other exclusion criteria) I am not seeing why they should be excluded. It would be good to say a little more about why this exclusion criteria was used.

Validity of the findings

No comment

Additional comments

The authors have addressed all my major concerns and I am happy to recommend the paper for publication.

I have made a few minor comments. However, most of my comments are language queries as I had small difficulties in certain points when reading through. I would recommend the authors give the paper one last proof-reading pass as I am sure there might be further minor things which I might have missed.

---

## Round 0.3 · accepted · Accept

It is my opinion that the authors have responded to all reviewer comments and that the final manuscript will make a nice contribution to the emerging literature on this topic.